# Controlling noise with self-organized resetting
Felix J. Meigel[1,2] & Steffen Rulands [1,2] ✉

Biological systems often consist of a small number of constituents and are therefore inherently noisy. To function effectively, these systems must employ mechanisms to constrain the accumulation of noise. Such mechanisms have been extensively studied and comprise the constraint by external forces, nonlinear interactions, or the resetting of the system to a predefined state. Here, we propose a fourth paradigm for noise constraint: self-organized resetting, where the resetting rate and position emerge from self-organization through time-discrete interactions. We study general properties of self-organized resetting systems using the paradigmatic example of cooperative resetting, where random pairs of Brownian particles are reset to their respective average. We demonstrate that such systems undergo a delocalization phase transition, separating regimes of constrained and unconstrained noise accumulation. Additionally, we show that systems with self-organized resetting can adapt to external forces and optimize search behavior for reaching target values. Self-organized resetting has various applications in nature and technology, which we demonstrate in the context of sexual interactions in fungi and spatial dispersion in shared mobility services. This work opens routes into the application of self-organized resetting across various systems in biology and technology.

Many non-equilibrium systems, particularly in the context of biology, consist of a small number of constituents and are inherently noisy[1–7]. Such systems perform specific functions, such as sensing and signal processing in cells[8–10], or the coordination of animal movements in flocks and herds[11,12]. Similarly, in technological applications, non-equilibrium systems are used in chemical reaction containers[13,14], or to coordinate shared mobility solutions[15]. If unregulated, noise tends to accumulate over time such that the behavior of such systems becomes increasingly unpredictable. To perform a function, these systems must therefore employ mechanisms to constrain the accumulation of noise.

These mechanisms for controlling noise have been extensively studied across disciplines, such as in biological[16–18] and technological[19–21] contexts: First, noise can be constrained by external fields that restrict the time evolution of a stochastic system to a phase-space region centered around a stable attractor. Specifically, these external fields generate generalized forces that suppress small perturbations away from the attractor, continuously reducing noise within the system. For example, during embryonic development, the determination of cell identity is controlled by external morphogen gradients, leading to spatially separated domains in the embryo[8,10,22,23]. In these systems, the coordination of constituents is regulated externally, without relying on interactions between the constituents.

In contrast to extrinsic forces, fluctuations can also be constrained intrinsically using self-organization: a stable steady-state emerges from interactions between constituents if the strength of these interactions is significantly greater than the amplitude of the noise. Examples of the self-organized constraint of noise are abundant in nature, ranging from flocking phenomena in active matter systems[2,11,16] to genetic switches in cells[24–26].

Instead of the continuous suppression of fluctuations by external forces or self-organization, noise can also be controlled by resetting the constituents of a system to predefined states. In these systems, noise is controlled by time-discrete jumps that reset the system to a predefined state. These processes are commonly referred to as stochastic resetting. Because the state, to which the system is reset to, is extrinsically defined, we refer to these mechanisms as extrinsic resetting. Extrinsic resetting leads to the emergence of stationary distributions with finite variance[27,28] such that dispersion and extreme fluctuations are constrained[29,30]. Extrinsic resetting processes may include drift-diffusion dynamics[31–33], multiplicative diffusion[34–36], and space-dependent resetting rates and positions as well as non-ergodic stochastic dynamics[36–42]. Extrinsic resetting events are, for example, realized by the RNA-Polymerase where misaligned polymer ends are randomly cleaved[43] or employed in the context of biochemical systems to grant efficiency of reaction dynamics[14,44–46].

[1]Max Planck Institute for the Physics of Complex Systems, Dresden, Germany. [2]Ludwigs-Maximilians-Universität München, Arnold Sommerfeld Center for Theoretical Physics, München, Germany. ✉e-mail: rulands@lmu.de

Here, we investigate a fourth class of mechanisms for the constraint of fluctuations in which system constituents are reset to positions determined in a self-organized manner by time-discrete interactions (Fig. 1a). Examples of such dynamics are ubiquitous in nature and describe, for example, the averaging of molecular concentrations during the fusion and fission of intracellular organelles and chromatin-domain droplets[47–51], or the stem-cell population dynamics during intestinal crypt fusion[52]. Further prominent examples include the genetic recombination dynamics during sexual reproduction[53,54], or the mixing of populations in ecological systems, in origin-of-live scenarios, and genetic tracing experiments[55–58].

We illustrate the rich dynamic behavior inherent to self-organized resetting schemes by focusing on the paradigmatic example of cooperative resetting. This example involves a many-particle system driven by two dynamics: the accumulation of noise governed by a stochastic process for each constituent, and random events of pair-wise resetting to their respective average (Fig. 1b). We show that self-organized, cooperative resetting leads to a stationary state with bounded variance if interactions are sufficiently long-ranged. In analogy to condensed-matter physics[59], we refer to the boundedness of the variance as localization. As interactions become increasingly short-ranged, self-organized resetting undergoes a delocalization transition. In the relevant case that interactions between particles decay quadratically, we identify a second-order phase transition depending on the particle density. As a consequence of localization, stochastic resetting enhances the search behavior of the stochastic process, resulting in a decreased time for constituents to reach a target state compared to Brownian particles. We exemplify the generality of our findings by using the framework of self-organized, cooperative resetting to explain the fitness advantage of sexual interaction in fungi and to demonstrate its applicability in designing organization strategies in shared mobility services.

## Results

### Definition of cooperative resetting

To investigate general properties of self-organized resetting systems, we focus on a simple, but paradigmatic model that comprises $N$ particles described by their positions $X_i$ in one spatial dimension. These particles undergo Brownian motion with a diffusion constant $D$. Pairs of particles can interact with a rate $\mu$ that depends on the distance $\delta = |X_i - X_j|$ between the particles. Upon such an interaction, the positions of both particles are set to their respective mean position, $(X_i + X_j)/2$. We use the term interactions to indicate that the dependence of the behavior of a given particle on other particles without implying an association with energy levels between the particles. Beyond a characteristic length scale $\delta_0$, these interactions decay algebraically with an exponent $\alpha$,

$$\mu(\delta) = \frac{\mu_0}{1 + (\delta/\delta_0)^\alpha}. \tag{1}$$

As we will discuss below, this choice of kernel allows us to draw conclusions about general interactions.

To describe the collective dynamics of this system, we study the single-particle probability density, $p(x, t)\mathrm{d}x$, of finding a particle between positions $x$ and $x + \mathrm{d}x$. The time evolution of $p(x, t)$ is governed by two processes: the effect of Brownian motion, which depends on the single-particle density $p(x, t)$, and the pair-wise resetting, which depends on the two-particle density $p_2(x, x', t)$. Using an operator notation, the time evolution of $p(x, t)$ follows an equation of the form

$$\partial_t p(x, t) = \hat{\mathcal{L}}[p(x, t)] + \hat{\mathcal{R}}[p_2(x, x', t)]. \tag{2}$$

We give the definitions of the operators $\hat{\mathcal{L}}$ and $\hat{\mathcal{R}}$ in the Supplementary Note 1.

We now ask under which conditions cooperative resetting can constrain the accumulation of fluctuations stemming from Brownian motion. Fluctuations are constrained if Eq. (2) admits a steady state with finite variance, i.e. it exhibits localization. We first note that cooperative resetting leads to a displacement of particles with an associated flux $-\partial_x \hat{J}_{\mathcal{R}} = \hat{\mathcal{R}}$. In the steady state, this flux must be balanced with the flux associated with Brownian motion, $-\partial_x \hat{J}_{\mathcal{L}} = \hat{\mathcal{L}}$, compare with Supplementary Note 2,

$$\hat{J}_{\mathcal{R}}[p_2(x, x', t)] = -\hat{J}_{\mathcal{L}}[p(x, t)]. \tag{3}$$

To assess if cooperative resetting admits for localized states, we test for the existence of a steady-state distribution that fulfills flux balance and is normalizable. To obtain a closed form in terms of the single-particle density, $p(x, t)$, we employ a mean-field approximation, $p_2(x, x', t) = p(x, t)p(x', t)$. We further approximate the left-hand side of Eq. (3) in the limit $x \to \infty$ to first order in $1/x$; see Supplementary Note 3 for the detailed derivation. With this, we obtain for $\delta$-distributed initial conditions at $x_0 = 0$ a flux-balance condition for localization in terms of the steady-state probability distribution $p_s(x)$,

$$\varrho \, p_s(\chi) \left| \frac{p_s(\chi)}{p'_s(\chi)} \right| \chi^{-\alpha} = -p'_s(\chi). \tag{4}$$

Here, we defined non-dimensional positions by rescaling lengths by the characteristic length scale of interactions $\chi = x/\delta_0$. We defined a non-dimensional parameter $\varrho = 2N\delta_0^2\mu_0/D$ that has the interpretation of a rescaled average particle density. $p'_s(\chi)$ denotes the derivative of $p_s(\chi)$ with respect to $\chi$. Note that due to the symmetry of the system, both sides of $p_s(\chi)$ must have an identical functional form for $|x| \to \infty$.

### Conditions for localization

In Eq. (4), we rephrased the resetting flux in a way that admits an intuitive interpretation. The first factor in the resetting flux, $p_s(\chi)$, is proportional to

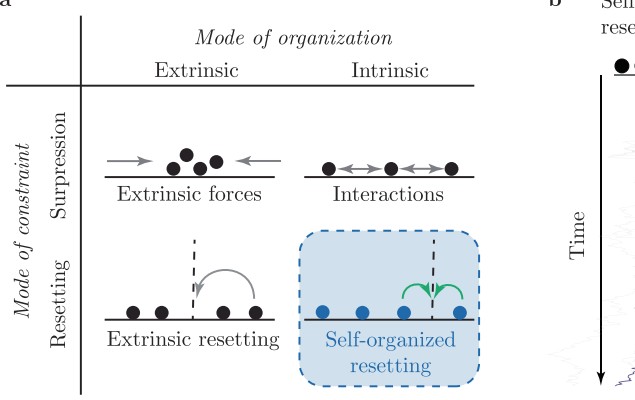

**Fig. 1 | Self-organized resetting as a mechanism for constraining fluctuations. a** Schematic illustrating different mechanisms that constrain the accumulation of noise. In the scheme, particles are depicted as filled circles, and external forces on the particles and interactions between the particles are depicted as arrows. The mechanism of self-organized resetting is emphasized by color-coding in blue and green. **b** Illustration of self-organized resetting: Exemplary trajectories obtained from agent-based simulations for cooperative resetting, where stochastic particles are randomly reset to their respective average. Each line represents the stochastic trajectory of a particle. Exemplary trajectories are highlighted by bold lines; resetting points are marked by circles. Random pair-wise resetting occurs with a rate of interaction $\mu$ which depends on the distance $\delta$ between pairs of particles.

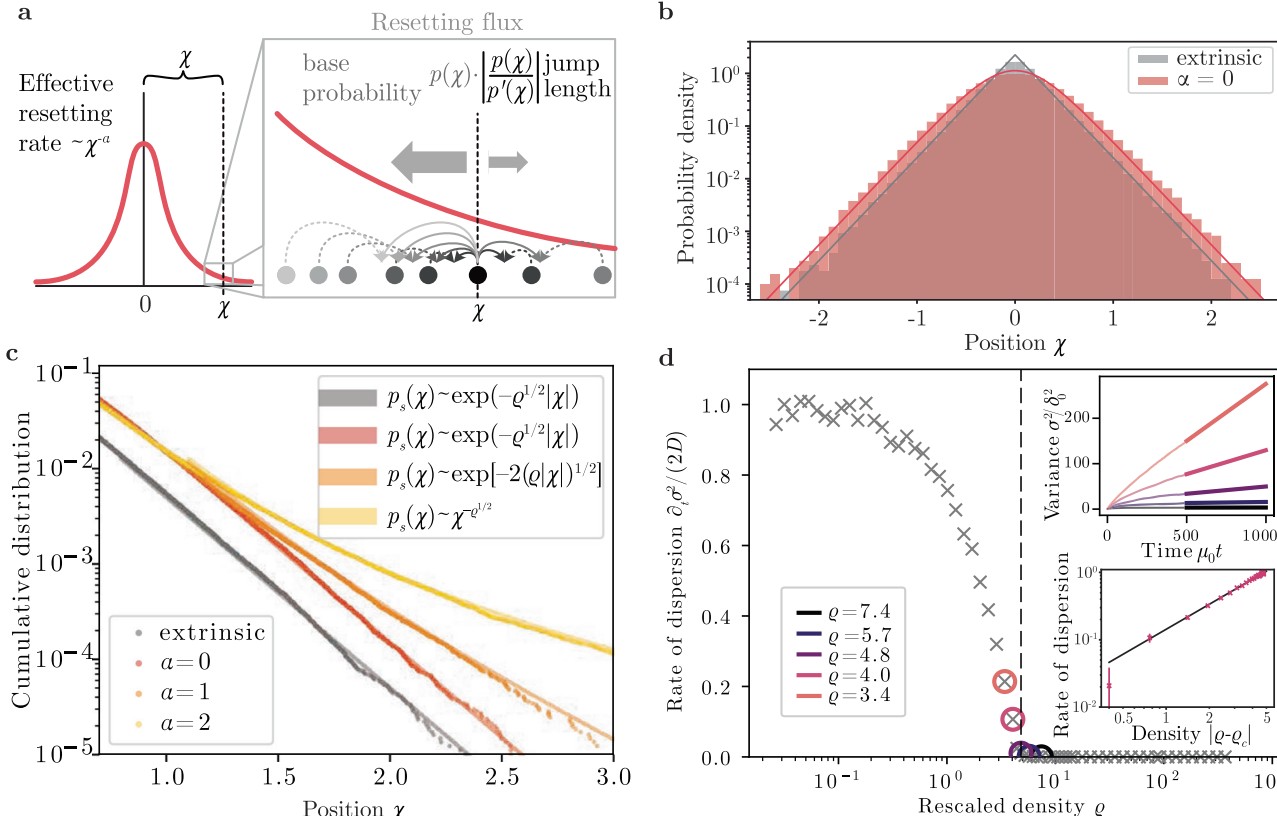

**Fig. 2 | Cooperative resetting exhibits a localization phase transition.**
**a** Interpretation of the approximated resetting flux in Eq. (4) as the product of an effective resetting rate, the density of particles contributing to the flux, and the effective jump length of resetting events. Particles are shown as filled circles and dotted arrows in different shades of gray indicate different possible cooperative resetting interactions. A probability density function is sketched in red. Filled-out bold arrows indicate the strength of probability fluxes due to resetting events. The dotted vertical line indicates the position $\chi$. **b, c** Agent-based simulations (dots) of $N = 2 \times 10^5$ particles for cooperative and extrinsic resetting with $\varrho = 21$. **b** Histograms for cooperative and extrinsic resetting which both decay exponentially for $\alpha = 0$. The histogram shows simulation data, the bold lines show the analytical result according to Eq. (4). Extrinsic resetting is depicted in gray; cooperative resetting is depicted in red. **c** The cumulative distribution is defined as $\int_x^\infty p_s(x)\,\mathrm{d}\,x$. Lines represent the predicted functional dependencies from Eq. (5). Simulation results are shown as dots. $\varrho$ is calculated from the simulation parameters, and the offset of the theoretical prediction is fitted to the simulation data. Extrinsic resetting

is shown in gray, and cooperative resetting is stratified in color for different values of $\alpha$. **d** For $\alpha = 2$ we find a phase transition dependent on the particle density. The top inset shows the increase in the variance over time. For particles performing Brownian motion, the variance increases as $\sigma^2 = 2Dt$. The slopes of the increase of the variance in the long-time limit (thick lines in inlay) define an effective order parameter, which is shown in the main plot. If the rescaled density is below a critical density, $\varrho < \varrho_c$, the probability density function delocalizes. Crosses depict results from agent-based simulations of $N = 2 \times 10^5$ particles. Circles depict values of $\varrho$ that are represented in the top inlay with matched color. The dashed line is the critical density $\sqrt{\varrho_c} = 2.3$ estimated analytically; compare with the Supplementary Note 4. The bottom inlay shows a power-law dependence of the rate of dispersion on rescaled density in the vicinity of the phase transition. The solid line is a nonlinear least-squares fit with an exponent $1.223 \pm 0.012$ (95% confidence intervals). Magenta crosses show the numerically measured rate of dispersion and error bars indicate the 95% confidence intervals.

the number of particles contributing to the flux at position $\chi$, the second factor gives the effective length of jumps due to resetting, and the third factor is the first-order approximation of the resetting rate of Eq. (1), Fig. 2a. Eq. (4) is a first-order nonlinear differential equation, which admits analytical solutions. Solving for $p_s(\chi)$ for given values of $\alpha$ and $\varrho$ then allows us to determine under which conditions cooperative resetting admits for localized solutions and to calculate the functional form of $p_s(\chi)$ in the limit $|\chi| \to \infty$.

Using Eq. (4), we now investigate localization conditions for different values of $\alpha$ and $\varrho$. For $\alpha = 0$, particles undergo pair-wise resetting independent of their distance. In this case, Eq. (4) admits stationary solutions and the cooperative resetting process exhibits localized states. Specifically, the stationary solutions asymptotically approach an exponential form, $p_s(\chi) \sim e^{-\sqrt{\varrho}|\chi|}$, for $|\chi| \to \infty$. In this case, cooperative resetting exhibits localization for all values of $\varrho$. We corroborated this result and all further results by agent-based simulations of Brownian motion and the resetting processes defined above. The stationary probability density function of the cooperative resetting process asymptotically matches that of the extrinsic

resetting process for $|\chi| \to \infty$, provided that the extrinsic resetting rate is set to $\mu_{\mathrm{ext}} = \mu_0 N$ and the resetting position is fixed to be the initial position $\chi_0$ (Fig. 2b). However, extrinsic resetting increases the likelihood of a particle being near the initial position $\chi_0$ compared to cooperative resetting, resulting in a higher probability in the tails of the density function for the cooperative resetting process.

If resetting is distance-dependent, $\alpha > 0$, the resetting rate depends on the local particle density. In this case, Eq. (4) yields for $\alpha \neq 2$ and in the limit $|\chi| \to \infty$ stationary solutions of exponential form,

$$p_s(\chi) \sim \exp\left[-\sqrt{\varrho}|\chi|^{1-\alpha/2}/(1-\alpha/2)\right]. \tag{5}$$

These solutions have decaying tails for $\alpha < 2$. In this case, resetting interactions are sufficiently long-ranged such that a balance between Brownian motion and resetting gives rise to a localized steady state (Fig. 2c). For $\alpha > 2$ the tails do not decay such that Eq. (4) does not admit steady-state solutions which are normalizable probability density functions. In this case, the

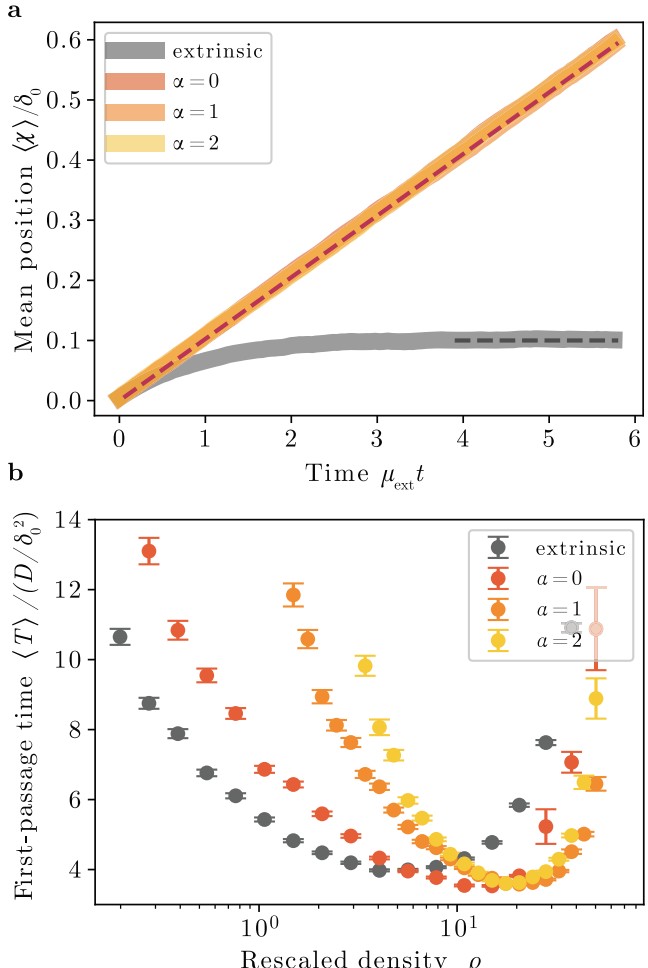

**Fig. 3 | Adaptation and search behavior of self-organized resetting. a** Localized states emerging from cooperative resetting differ qualitatively from localized states emerging from extrinsic resetting in their response to external forces and search behavior. Agent-based simulations ($\varrho = 21$) for cooperative resetting and extrinsic resetting which are subject to a fixed drift $v\delta_0/\mu_{ext} = 0.1$. For extrinsic resetting, the localized states remain close to the resetting position $\chi_0$, while for cooperative resetting the localized state translocates with the drift $v$. **b** The minimal mean first-passage time for a marked particle in an ensemble of 200 particles to reach a target at a distance $l = 0.4$ is compared for intrinsic and cooperative resetting as $\varrho$ is varied through $\mu_0$. The minimal mean first-passage time is reduced for the cooperative resetting scheme. Averages are taken over $n = 5000$ simulations and error bars represent the standard error of the mean.

stochastic dynamics is dominated by Brownian motion. Therefore, cooperative resetting exhibits a delocalization transition for increasing values of $\alpha$: Localization occurs for all positive values of the rescaled density $\varrho$ for $\alpha < 2$ and no localization occurs for $\alpha > 2$. As a consequence, we can make predictions for general interaction kernels: If the integral $\int d\delta\, \delta\mu(\delta)$ exists, i.e., the interaction kernel decays faster than $\mu(\delta) \sim \delta^{-2}$ in the limit $\delta \to \infty$, it does not admit for localization. This in particular includes interaction kernels with any exponential decay.

Interactions decaying with an exponent of $\alpha = 2$ are ubiquitous in nature (This applies particularly for three-dimensional systems, to which our results straight-forwardly extend.). They also form a special case in the cooperative resetting process as the exponent Eq. (5) diverges. In this case, the steady-state solution of Eq. (4) decays algebraically, $p_s(\chi) \sim \chi^{-\sqrt{\varrho}}$, which we also find in numerical simulations (Fig. 2b). The probability density function is not normalizable for $\sqrt{\varrho} < 1$. This suggests that for $\alpha = 2$ the cooperative resetting process undergoes a delocalization transition with decreasing values of the rescaled density, $\varrho$. For $\varrho \to 0$ the typical distance

between particles is large such that the overall rate of resetting is small. This regime is therefore dominated by diffusion. For $\varrho \to \infty$, resetting processes dominate and we find that the probability density localizes (Fig. 2d). While the normalization condition gives a rough estimate for the critical value of the rescaled density of $\varrho_c \approx 1$, an analysis of Eq. (4) including higher orders yields an improved approximation of $\sqrt{\varrho_c} \approx 2.3$, compare with the Supplementary Note 4. In general, these results imply that with an increasing number of particles, $N$, or a decreasing diffusion constant, $D$, cooperative resetting with quadratically decaying interactions exhibits a localization phase transition.

## Adaptation and search behavior

Having studied the steady-state behavior of the cooperative resetting process, we now ask how cooperative resetting systems respond to extrinsic forces. To this end, we apply a constant force $F$ and study the response to it in the cooperative resetting model. We compared our results to the extrinsic resetting model. In the over-damped limit, the constant force gives rise to a constant velocity of particles, $v$, which is proportional to $F$. For the extrinsic resetting scheme the localized state remains close to $\chi_0$ and assumes a skewed distribution. The mean position takes a constant value that deviates from the resetting position, $\langle \chi \rangle - \chi_0 = v/\mu_{ext}$[60]. For the cooperative resetting scheme, the extrinsic force yields an additional drift flux of the form $\hat{J}_{\mathcal{F}} = vp(\chi, t)$ in Eq. (3). Making the Galilean transformation $\chi' = \chi - vt$ this flux vanishes. For cooperative resetting, the localized states remain symmetric and we instead find that the system responds with a translocation with constant velocity to the extrinsic force (Fig. 3a). Therefore, cooperative resetting systems can adapt to extrinsic forces while at the same time constraining intrinsic noise.

Stochastic resetting processes have been intensively studied in the context of search problems, in which one is interested in the time a particle takes to reach a given target position for the first time. Extrinsic stochastic resetting processes have the counter-intuitive property that they exhibit lower mean first-passage times compared to pure Brownian motion[29,60]. In extrinsic resetting, an optimal resetting rate minimizes first-passage time for any target distance $l = \chi^* - \chi_0$: high resetting rates lead to frequent resets, even for particles close to targets, while low resetting rates result in infrequent resets, allowing the particle to wander off in the wrong direction.

We numerically test, whether this hallmark of extrinsic resetting is also present in cooperative resetting. To this end, we study the first-passage time of a single, randomly chosen particle. Our findings align with the well-known behavior of extrinsic resetting, indicating that cooperative resetting similarly achieves an optimal state that minimizes mean first-passage time for a given target distance. Specifically, we numerically find that if the cooperative resetting dynamics admits for a localized state, there is an optimal density $\varrho^*$ which minimizes the first-passage time for a given target distance, $l$, (Fig. 3b). The search is generally improved for cooperative resetting as the minimal mean first-passage time is lower compared to extrinsic resetting. This improvement highlights a key difference: cooperative resetting reduces the likelihood of particles remaining near the initial position, leading to enhanced flux toward the target. This is intuitive, as in cooperative resetting, the reduced probability of the particle being close to the initial position $\chi_0$ (Fig. 2b) leads to an increased flux away from $\chi_0$. This reduces the impact of retracting steps close to the target. A similar effect is achieved in models comprising a non-resetting window around $\chi_0$, which has been shown to reduce the minimal first-passage time for extrinsic resetting[29]. Next, we apply our theoretical findings to the inhibition of fluctuations in biological and technical systems.

## Sexual recombination in fungi

We illustrate the application of self-organized resetting through the paradigmatic biological example of sexual reproduction. Specifically, we focus on the sexual interactions among filamentous fungi: Coenocytic fungal species, such as *Zygomycota* and *Glomeromycota*, are multi-nucleated organisms containing nuclei with diverse genetic identities[61–65]. These filamentous fungal species form networks of interconnected hyphae with nuclei that

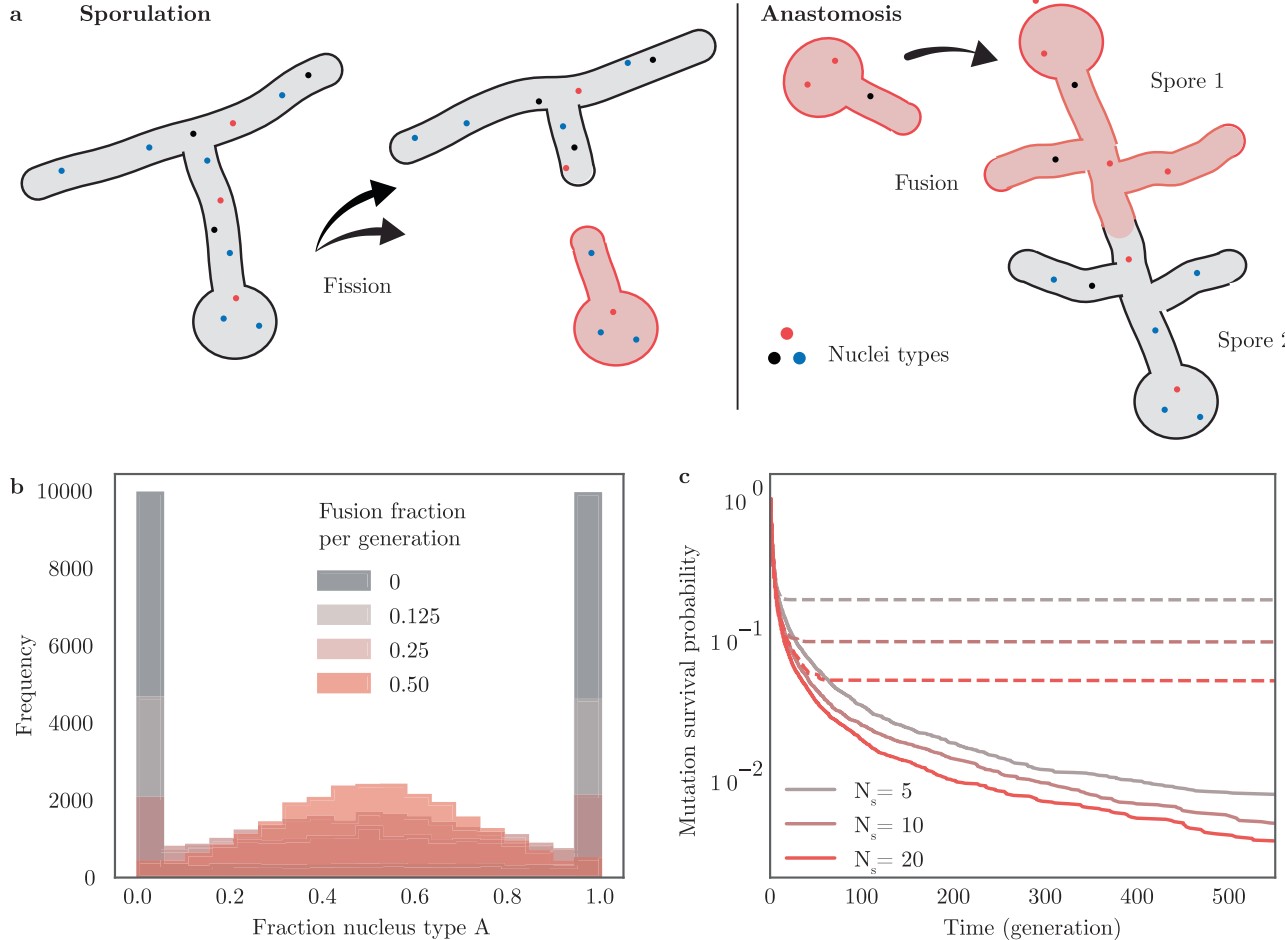

**Fig. 4 | Fungi constrain genetic variability by self-organized resetting. a** Multi-nucleated fungi undergo cycles of sporulation and anastomosis, as the hyphae of different fungi fuse and allow the exchange of nuclei. Fungi spores carry only a subset of nuclei and sporulation acts effectively as a fission event. Iterative fusion and fission implement pair-wise cooperative resetting. **b** Stochastic simulations of fungal populations demonstrate hyphal fusion to preserve genetic heterogeneity of neutral genes by self-organized resetting. Genetic heterogeneity is quantified in a population snapshot after 25 generations. Different colors represent different probabilities for fungi to undergo anastomosis with a different fungus after sporulation and subsequent germination. The spore size is $N_s = 20$. **c** Simulations show an increased probability for fixation of neutral mutations in the absence of anastomosis (dotted lines). Fungi undergo anastomosis with a probability of 25% after germination (bold lines). Different colors indicate different effective spore sizes in terms of the number of enclosed nuclei.

freely move in the cytoplasm. The nuclei in a fungus replicate asexually through division during colony growth. A fungal organism as a whole reproduces via sporulation, during which a subset of nuclei is encapsulated and emitted from the fungus. Upon germination, these spores establish new fungal colonies. Hyphae from different fungi can fuse upon contact (anastomosis), facilitating the exchange of nuclei between the two fungi (Fig. 4a). The genetic diversity of nuclei is hypothesized to grant the ability of fungi to rapidly evolve in changing environments[61].

The number of nuclei per spore is relatively small compared to the total number of nuclei in a fungal network and varies from a few tens to thousands of nuclei in a single spore[62]. On the level of an isolated fungus, the homogenous states are absorbing states, where a single nuclei genotype can achieve fixation. One would therefore expect genotypes within a fungus to go extinct on the timescale of a few fungal generations. In contrast, hyphal fusion leads to an effective averaging of the fractions of different genotypic nuclei among fungi. This is a manifestation of cooperative resetting. We therefore predict that the localization of the probability density function of nuclear genotype fractions could lead to long-term stabilization of genetic diversity.

To test this, we performed stochastic simulations of the numbers $n_i$ of nuclei with a genotype $i$ in a population of $N = 2.5 \times 10^4$ fungi, accounting for fungal growth with nuclei replication dynamics as well as sporulation, and anastomosis. Here, a cycle of germination followed by sporulation

defines a unit of time which we refer to as a generation. In a minimal model, we consider two genotypes with equal fitness, a constant generation progression time, and restrict hyphal fusion to fungi colonies in the same generation. We set our model parameter based on physiological estimates: In agreement with ref. 62, we varied the number of nuclei of spores between $N_s = 5$ and 20. We fixed the number of fungi by setting the number of germinating spores to one per colony, considering that the successful germination rate of fungi in wildlife conditions is significantly below the rates measured in the laboratory[63]. We varied the number of successful anastomosis events between germinated spores in the range of 12%–50%[61,63,65]. After germination, our model comprises a growth phase of 7 nuclear divisions, where the set of newly formed nuclei stems from a one-step Moran process. For sporulation, a random subset of nuclei is picked from the fungal colony. An equal number of nuclei of both genotypes was set as an initial condition.

Our simulations show that in the absence of fungal fusion, genetic diversity is largely lost over the course of the simulation (25 generations), as more than 70% of colonies exhibit fixation of one type of nuclei. Sexual recombination by hyphal fusion leads to a localization of the fraction of genotypes at 50% (Fig. 4b). While cooperative resetting facilitates the maintenance of existing genetic diversity, it also leads to the extinction of new mutations (Fig. 4c). By maintaining genetic diversity and thus granting adaptability while simultaneously hindering the frequent fixation of

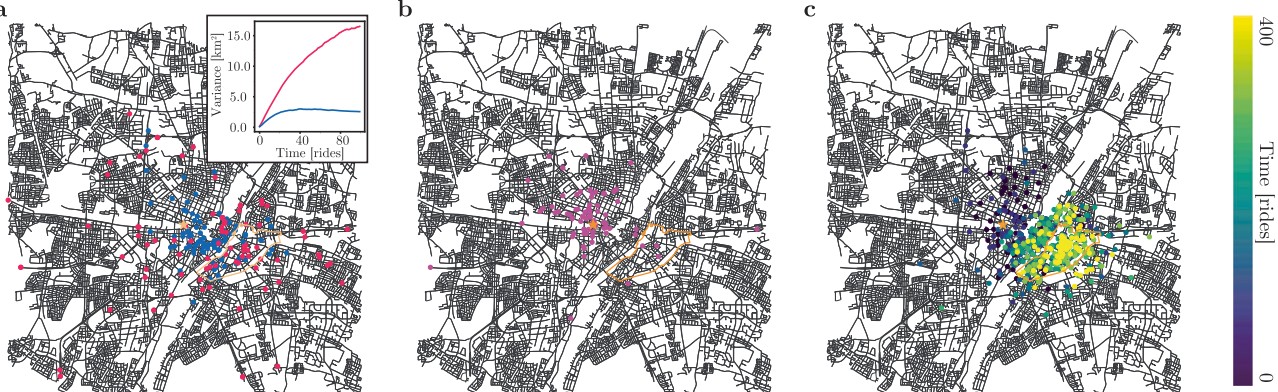

**Fig. 5 | Cooperative resetting constrains the spatial dispersion of vehicles.**
**a** Simulation snapshot of the spreading dynamics of a vehicle cohort ($N = 100$) in the absence of resetting (red) and while undergoing cooperative resetting (blue) after 100 random rides. Vehicles perform random walks on the street network of Munich. Each ride has a length of 10 steps and 10% of rides were followed by a cooperative resetting event. The initial position (central station of Munich) is marked by an orange star. The inset shows the spatial variance of the vehicle cohort over time. In

the presence of cooperative resetting, the spatial variance is bounded (localization). **b** Simulation snapshot of a scooter cohort (pink) subject to an extrinsic resetting scheme to the initial position (orange star) after 400 rides. Preferential rides (1% of the rides) to the quarter `Au-Haidhausen' (orange outline) do not lead to re-localization in this quarter. **c** Same as (**b**), but for cooperative resetting (10% of rides). An overlay of different time points (color-coded) demonstrates the temporal adaptation as the spatially localized cohort shifts to the quarter Au-Haidhausen.

mutations and thus maintaining robustness, cooperative resetting helps to stabilize the trade-off in the adaptability-robustness dilemma[66–68]. Significantly, our qualitative findings are robust to changes in the parameter choice, as long as the number of nuclei in a spore is much smaller than the number of nuclei in the fungal colony.

### Localization of shared mobility devices
As an illustration of utilizing cooperative resetting in a technical application, we study the dispersion of vehicles in shared mobility services. Users of these services book vehicles for individual rides, where they pick up the shared vehicle close to their starting position and drop it off at their destination. The major challenge for companies offering shared mobility services is providing vehicles at locations with a high demand[69]. At the same time, the randomness in pick-up and drop-off gives rise to a spatial dispersion of vehicles over time,[15], resulting in an increase in the average distance between vehicles, which consequently implies a reduction in the local availability of vehicles. Thus, mechanisms that constrain the spatial dispersion of vehicles are of central interest to companies providing shared mobility services.

Next, we exemplify that cooperative pair-wise resetting is a possible viable mechanism to implement a self-organized resetting scheme for shared mobility services. We simulate simple pick-up and drop-off dynamics of the shared vehicles as a random walk on the street network of Munich, which we obtained from Open Street Map using the `python` package `osmnx`[70] and in which intersections define nodes. We set the length of a random ride to $N_{length} = 10$ streets being traversed. We compared three models: First, an implementation without resetting in which vehicles may freely disperse; Second, a model with extrinsic resetting, in which vehicles are reset to the initial starting point. This mimics services in which vehicles are collected and collectively dropped off at central locations. Thirdly, we also study a model, in which vehicles are reset pair-wise through the coordinated interactions of users. In the simulations, this means that we pick two random scooters and reset them to the junction closest to their spatial average.

Expectedly, without resetting, vehicles disperse unconstrained, but subdiffusively while extrinsic or cooperative resetting constrains the dispersion of vehicles to a defined area (Fig. 5a, b). We next study the effect of a change in customer preferences by assuming that, on average, one in 100 rides has a specific destination on the map. Because cooperative resetting allows for the simultaneous constraint of noise and an adaption to extrinsic forces (Fig. 3a) we expect that a cooperative resetting scheme will adapt to changes in customer preferences while restricting the dispersion of vehicles. With extrinsic resetting of the vehicles, the probability density function remains localized around the resetting position (Fig. 5b) while for

cooperative resetting the mode of the probability density function adapts to the bias in the drop-off locations (Fig. 5c).

## Discussion
With self-organized resetting, we have studied a paradigm for the constraint of fluctuations in stochastic systems. We showed that self-organized resetting allows for the suppression of the accumulation of noise in stochastic many-particle systems. This mechanism is conceptually different compared to other mechanisms for the constraint of noise using external signals, or self-organization schemes that demand the continuous suppression of noise. We exemplify the concept of self-organized resetting by focusing on the mechanism of cooperative resetting, where random pairs of constituents reset to their respective average. We showed that such a self-organized resetting scheme gives rise to localization phase transitions as a function of the spatial decay of resetting interactions and the particle density. Cooperative resetting systems respond differently to external forces if localization occurs and show improved search behavior compared to extrinsic resetting. Our work shows that the localization and search properties of extrinsic resetting schemes can be obtained in a self-organized manner without the need for extrinsic control.

While studying the localization phase transition in a simple cooperative resetting scheme for Brownian noise admits for analytical solutions, we expect our results to be qualitatively valid also for multiplicative and colored noise. Making the ansatz of the resetting flux balancing a flux attributable to the spreading dynamics as in Eq. (3), our calculations are generalizable to any stochastic system that is describable in terms of a moment expansion to second order. Additionally, our approach can be extended to include additional extrinsic force fields or to higher dimensions.

For search behavior, our simulations show that self-organized cooperative resetting replicates a key feature of extrinsic resetting: the existence of an optimal resetting rate for a given search target distance. Our results suggest that optimized first-passage times outperform those of extrinsic resetting, with an optimal resetting rate persisting in cooperative resetting only when localization occurs. Exploring whether the optimal resetting rate exhibits phase transition features, as seen in extrinsic resetting with moving absorbing boundaries[71], could enhance our understanding of optimizing multi-agent search tasks.

We have demonstrated two applications of self-organized resetting: sexual interactions in filamentous fungi and the spreading dynamics of shared mobility solutions. While we here focused on two specific examples, we expect that self-organized resetting is a general and abundant mechanism for constraining fluctuations. Further applications for self-organized

resetting range from intra-cellular signaling pathways on fusion and fission organelles, stem-cell dynamics in fusing intestinal crypts, or - as an economical application - the redistribution of wealth through altruistic donations in economic systems[72,73].

By focusing on the conditions for a localized state, we did not discuss the transient dynamics of approaching such states. In extrinsic resetting, such transients are associated with distinct dynamical phase transitions[74]. Our framework could be expanded to include non-Markovian dynamics, where particles interact with their own history as has been studied for single particles in ref. 75. While such systems do not comprise many-particle dynamics they share with self-organized resetting schemes that the resetting mechanism arises intrinsically from the stochastic process. Incorporating additional memory effects into self-organized resetting could unveil further counter-intuitive collective phenomena. Overall, we view self-organized resetting as a fundamental and versatile mechanism for controlling noise accumulation in stochastic systems with many constituents, offering broad applicability.

## Methods

### Numerical routines and simulation

For the numerical validation backing our general analytical findings, we performed agent-based simulations in `python 3.8`. For the Brownian motion, we perform independent simulations of particles by the numerical integration of stochastic differential equations. This yields a set of stochastic realizations. To implement resetting, at each time step, we compute the rate of each pair of particles to reset to their respective mean position. Based on this rate, we update two randomly chosen particles by setting them to their mean positions using a stochastic simulation algorithm. In between resetting events, the particles again independently perform Brownian motion. Statistical analysis was performed in `python 3.8`.

## Data availability

The simulation data (.txt-files) and source data underlying this work are available in the Zelondo repository under https://doi.org/10.5281/zenodo.14274930.

## Code availability

Simulation routines are described in the "Method" section. Code snippets are available from the corresponding author upon reasonable request.

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

## Acknowledgements

We thank M. Ciarchi, R. Mukhamadiarov, and F. Jülicher for their helpful feedback and the entire Rulands group for critical discussions. This project has received funding from the European Research Council (ERC) under the European Union's Horizon 2020 research and innovation program (grant agreement no. 950349).

## Author contributions

F.J.M. conducted the research and performed the data analysis. S.R. supervised the research. Both authors wrote, reviewed, and approved the final manuscript.

## Funding

## Competing interests

The authors declare no competing interests.
