## [Transparent Peer Review file · Communications Physics]

Controlling noise with self-organized resetting

Corresponding Author: Professor Steffen Rulands

This manuscript has been previously reviewed at another journal. This document only contains information relating to versions considered at Communications Physics.

Version 0:

Reviewer comments:

Reviewer #1

(Remarks to the Author)

The paper considers general principles of how to control the build-up of intrinsic noise in a system of a finite number of constituents. The build-up of noise must be controlled in order for the system to accurately perform some collective function. In the introduction three mechanisms, existing in the literature, to control noise are identified, and a fourth mechanism is proposed.

The proposed mechanism is a variation of the idea of stochastic resetting. The idea and effects of resetting have been of considerable interest in recent years in a number of contexts, and resetting now forms a subfield of nonequilibrium statistical physics.

The novel resetting mechanism that is proposed is simply to stochastically reset pairs of particles to their average position. It is shown by numerical simulations and a mean field theory that this mechanism will result in a localised distribution of the constituent particles, if the interactions are sufficiently long-ranged. The mean-field theory reveals that different spatial distributions of the particles are possible according to the details of the interaction kernel. Thus, a simple idea leads to rich results. Applications to genetic diversity in fungal fusion/fission dynamics and control of vehicular spread in shared mobility services are discussed. These applications appear to be of interest (although I am not expert in these areas)--they illustrate the general applicability of the proposed mechanism. Therefore I believe the paper is a worthwhile addition to the field of stochastic resetting and will be of interest to a broad community.

The paper is generally clearly written and accessible to the non-expert. A few suggestions for improvement are made below.

1. To make Figure 1 self-contained it would be helpful if μ were defined in the caption as the rate of interaction between a pair of particles separated by distance δ .
2. The operator appearing in Equation (2) seems to be incorrectly defined in Appendix A equation (A1), where the diffusive operator should involve second derivative with respect to space (not time).
3. on p.3, column 2, the second last paragraph beginning 'The stationary solution...' is rather opaque. The meaning of this paragraph could be made clearer.
4. p.4 last paragraph. The second last sentence 'A similar effect ...' cites reference [37] but this probably should be reference [29]. The authors should check.
5. p.7 third paragraph. Following the sentence 'While cooperative resetting facilitates the maintenance of existing genetic diversity, it also leads to the rapid extinction of new mutations', some comment could be made on the significance of this observation e.g. it could be made clear whether eliminating neutral mutations is desirable.
6. p.7 before Discussion section, a Figure reference (Fig ??b) should be corrected.

Reviewer #2

(Remarks to the Author)

The Authors consider an interesting problem of self-organized resetting mechanism in stochastic process. The manuscript is well written and of interest to the scholars working on random processes under resetting and their applications. The considered model is applied to a system of sexual interactions in fungi and spatial dispersion in shared mobility services. Therefore, the present manuscript is of interest to wider community of researchers.

Here, I give some additional comments on the manuscript.

- Introduction: When mentioning the “non-ergodic stochastic dynamics [37–42]”, Ref. [36] also deals with non-ergodicity of stochastic process with resetting (first work on non-ergodicity of geometric Brownian motion under resetting).
- The transition to the non-equilibrium stationary state, i.e., the emergence of dynamical phase transition should be explained in more details. Some comparison with the known results for the transition to the non-equilibrium stationary states in case of a Brownian motion under resetting with and without constant external force and other more generalized systems, which are discussed in many of the cited references of the manuscript (and, see also [Phys. Rev. E 91, 052131 (2015)], [Phys. Rev. E 108, L052106 (2023)]), could be shortly discussed/compared in order to understand the corresponding results in case of self-organized resetting.
- Section “Results” - “Adaptation and search behavior”: The results for the mean first passage times should be explained in more details in order to be clear the comparison to the well-known results in the literature for a Brownian search under resetting or some more generalized search processes under resetting.
- Can we give a general conclusions from the obtained results in the manuscript for a general form of the decay of interactions $\mu(\delta)$, for example if mean and/or variance of $\mu(\delta)$ in Eq. (1) exist or not?
- In the last paragraph before DISCUSSION there is a typo “...(Fig. ??b)”. Please correct it.
- Appendix A, relation A1: The partial derivative should be in respect to x . Please correct it.

Reviewer #3

(Remarks to the Author)

In this manuscript, the authors introduce and study a model of "self-organised" resetting which they study (partly) analytically and numerically. In this model of independent Brownian walkers a pair of walkers is reset to its center of mass with a rate that depends on the mutual distance between them. The authors mainly study a one-dimensional situation where the rate is an algebraically decaying function of the inter-particle distance with an exponent α . Interestingly they find markedly different behaviors for $\alpha < 2$ and $\alpha > 2$. For $\alpha > 2$, the diffusion is a stronger than the resetting mechanism and the system does not reach a steady state, while for $\alpha < 2$ there exists a normalisable steady state, where the density profile has a stretched exponential tail. The authors talk about a "localization" transition. Exactly at $\alpha = 2$, they find a similar transition as the density of particles ϱ crosses a critical value $\varrho = \varrho_c$, separating a delocalised phase at low density from a localised one at high density. The authors also study (numerically) the mean first passage time to a target and show that the search is generally improved by self-organized/cooperative resetting compared to standard/extrinsic resetting. Two applications of these concepts are discussed: one in a toy model of sexual interaction in fungi and the other in the context of shared mobility service design.

I find the model quite interesting (although the idea of "self-organized resetting" is not entirely new, see below), leading to a rich phenomenology, despite its simplicity. The paper is well written and I also appreciated that the authors provide a study which combines both analytical (although approximate) arguments, which seem quite reliable, and numerical simulations. In particular, the existence of this localization/delocalization transition is quite interesting and it will certainly trigger further research around such models. Therefore, I would like to recommend the publication of the present manuscript. I would like however to ask the authors to consider the following points:

1) As I said, a similar idea of "self-organized" resetting already appeared in the recent literature, for instance:

Majumdar, S. N., Sabhapandit, S., & Schehr, G. (2015). Random walk with random resetting to the maximum position. *Physical Review E*, 92(5), 052126.

Here also, the location of the resetting position is generated by the dynamics and not by an external operator.

2) The authors talk about μ as an interaction, which is a bit misleading, since it is not an interaction in the usual sense in statistical physics (i.e. associated to some energy for instance). This is a rate at which two particles "react". This could be emphasized a bit better (it is said in the present version but a bit hidden I think).

3) Could the author justify a bit more the mean-field approximation they use to solve (3). Does it become exact in the limit $N \rightarrow \infty$? This can be checked numerically for instance. Note that such decoupling is often used in the "hydrodynamic" description of many-particle systems (for instance in the context of the Dean-Kawasaki which the authors could refer to).

4) On p. 3, in Eq. (4) there should be a minus sign on the right hand side.

5) On p. 3, second column: the third paragraph should be part of the second one -- not a new one. As it is, it is a bit misleading because the authors still discuss the $\alpha = 0$ case there.

6) On p. 4, I guess that the authors mean that the density is not normalisable ** for $\rho < 1$ ** (and not the converse).

7) On p. 4, they use once ρ but use otherwise $\bar{\rho}$ to denote the mean density.

8) On p. 4, can one estimate how the rate of dispersion behaves close to $\bar{\rho}_c$. For $\rho < \bar{\rho}_c$, as ρ approaches $\bar{\rho}_c$, it probably vanishes as $\propto (\bar{\rho}_c - \rho)^\nu$ with some exponent ν . What is this exponent? Does the mean-field prediction gives the correct value?

Version 1:

Reviewer comments:

Reviewer #1

(Remarks to the Author)

The points I raised in my report have been satisfactorily addressed in the revised submission. I consider the paper suitable for publication.

Reviewer #2

(Remarks to the Author)

The Authors have addressed my comments and the manuscript can be accepted for publication.

Reviewer #3

(Remarks to the Author)

In this revised version the authors have satisfactorily answered to my comments and made the appropriate modifications. I am thus happy to recommend the publication of the present manuscript in Nature, Communications Physics.

We thank all reviewers for their appreciation of our work and the suggestions to further improve it. We have implemented all suggestions and marked them in blue in the revised manuscript.

Reviewer #1 (Remarks to the Author):

The paper considers general principles of how to control the build-up of intrinsic noise in a system of a finite number of constituents. The build-up of noise must be controlled in order for the system to accurately perform some collective function. In the introduction three mechanisms, existing in the literature, to control noise are identified, and a fourth mechanism is proposed.

The proposed mechanism is a variation of the idea of stochastic resetting. The idea and effects of resetting have been of considerable interest in recent years in a number of contexts, and resetting now forms a subfield of nonequilibrium statistical physics.

The novel resetting mechanism that is proposed is simply to stochastically reset pairs of particle to their average position. It is shown by numerical simulations and a mean field theory that this mechanism will result in a localised distribution of the constituent particles, if the interactions are sufficiently long-ranged. The mean-field theory reveals that different spatial distributions of the particles of the particles are possible according to the details of the interaction kernel. Thus, a simple idea leads to rich results. Applications to genetic diversity in fungal fusion/fission dynamics and control of vehicular spread in shared mobility services are discussed. These applications appear to be of interest (although I am not expert in these areas)---they illustrate the general applicability of the proposed mechanism. Therefore I believe the paper is a worthwhile addition to the field of stochastic resetting and will be of interest to a broad community.

The paper is generally clearly written and accessible to the non-expert. A few suggestions for improvement are made below.

We thank the reviewer for their appreciation of our work.

1. To make Figure 1 self-contained it would be helpful if μ were defined in the caption as the rate of interaction between a pair of particles separated by distance δ .

We agree with the reviewer that this information increases the clarity of Figure 1. We made the suggested edits to the caption.

2. The operator appearing in Equation (2) seems to be incorrectly defined in Appendix A equation (A1), where the diffusive operator should involve second derivative with respect to space (not time).

We thank the attentive reviewer for spotting this typo in the supplement. This typo is corrected in the updated manuscript.

3. on p.3, column 2, the second last paragraph beginning 'The stationary solution...' is rather opaque. The meaning of this paragraph could be made clearer.

We agree with the reviewer that this paragraph lacked clarity. We improved the wording in the paragraph to better communicate the comparison to extrinsic resetting dynamics. Also, regarding comment 5 of reviewer #3, we merged this paragraph with the previous paragraph.

Note that the merging of the paragraphs is not visible in the PDF file showing the differences between the original and the revised manuscript due to a limitation in the program diffTex that we used to generate this file. However, this change will become visible in the final, unannotated version of the manuscript.

4. p.4 last paragraph. The second last sentence 'A similar effect ...' cites reference [37] but this probably should be reference [29]. The authors should check.

We thank the reviewer for the careful reading of our manuscript and for pointing out this mistake. Indeed, this reference should be [29] instead of [37]. We have corrected this in the revised manuscript.

5. p.7 third paragraph. Following the sentence 'While cooperative resetting facilitates the maintenance of existing genetic diversity, it also leads to the rapid extinction of new mutations', some comment could be made on the significance of this observation e.g. it could be made clear whether eliminating neutral mutations is desirable.

We thank the reviewer for highlighting the importance of the biological interpretation of neutral mutations in the context of population fitness. The quoted sentence from the manuscript regards an evolutionary dilemma: Organisms need the ability to adapt to changing environments, which requires generating and maintaining genetic diversity through mutations, horizontal gene transfer, and other mechanisms. At the same time, they must preserve a functional, optimized genome to avoid the potentially harmful effects of excessive or maladaptive mutations. Genetic diversity is crucial because it enables bacterial populations to explore adaptive peaks in dynamic environments. However, preventing the fixation of mutations—particularly those that are neutral or

slightly deleterious—supports genomic stability. This is also elaborated on, for example, in the following references: [Takahashi et al, Balanced genetic diversity improves population fitness, *Proceedings of the Royal Society B: Biological Sciences* 285, 20172045 (2018) ; Grenier et al., Phenotypic Plasticity and Selection: Nonexclusive Mechanisms of Adaptation, *Scientifica* 2016, 7021701 (2016); Wagner, Robustness, evolvability, and neutrality, *FEBS Letters* 579, 1772–1778 (2005)]

In our manuscript, we demonstrate how self-organized resetting can help to stabilize this trade-off between preserving the genetic variability and hindering the frequent fixation of new mutations. As suggested, we now explicitly reference the adaptability-robustness dilemma in our manuscript and quote to the above mentioned references. The discussion is added at the end of the section “Sexual recombination in fungi”, page 6, right column in last paragraph.

6. p.7 before Discussion section, a Figure reference (Fig ??b) should be corrected.

We corrected this.

Reviewer #2 (Remarks to the Author):

The Authors consider an interesting problem of self-organized resetting mechanism in stochastic process. The manuscript is well written and of interest to the scholars working on random processes under resetting and their applications. The considered model is applied to a system of sexual interactions in fungi and spatial dispersion in shared mobility services. Therefore, the present manuscript is of interest to wider community of researchers.

We thank the reviewer for their appreciation of our work and their helpful suggestions.

Here, I give some additional comments on the manuscript.

- Introduction: When mentioning the “non-ergodic stochastic dynamics [37–42]”, Ref. [36] also deals with non-ergodicity of stochastic process with resetting (first work on non-ergodicity of geometric Brownian motion under resetting).

We thank the reviewer for pointing out that indeed Ref. 36 should correctly be cited also when referring to studying non-ergodicity of resetting processes in the current literature. We now cite Ref. 36 in the sentence about non-ergodicity.

- The transition to the non-equilibrium stationary state, i.e., the emergence of dynamical phase transition should be explained in more details. Some comparison with the known results for the transition to the non-equilibrium stationary states in case of a Brownian motion under resetting with and without constant external force and other more generalized systems, which are discussed in many of the cited references of the manuscript (and, see also [Phys. Rev. E 91, 052131 (2015)], [Phys. Rev. E 108, L052106 (2023)]), could be shortly discussed/compared in order to understand the corresponding results in case of self-organized resetting.

We agree that placing our results in context with related studies is valuable for understanding the broader implications of our work. The references provided are particularly relevant to extrinsic resetting: [Phys. Rev. E 91, 052131 (2015)] explores the transient dynamics and convergence to a non-equilibrium steady state in terms of a dynamical phase transition, while [Phys. Rev. E 108, L052106 (2023)] discusses a phase transition in the optimal resetting rate as a function of the distance to a moving absorbing boundary.

In our work, we focus on a fundamentally different phenomenon: a localization phase transition arising in cooperative self-organized resetting schemes. Unlike the dynamical phase transitions described in extrinsic resetting, the localization transition in our study results from the interplay of cooperative interactions and self-organized resetting. This leads to a distinct mechanism and set of outcomes that are not directly comparable to the extrinsic resetting scenarios discussed in the cited works.

Nonetheless, we find it noteworthy to point out that there is a broad variety of dynamical phases generally present in the context of extrinsic resetting. Studying the transient dynamics of approaching the non-equilibrium steady state or analysing our system in the context of absorbing boundary conditions might show new rich behaviour and additional dynamical phase transition dynamics.

To address the reviewer's concern, we have included a brief comparison of the underlying mechanisms and their distinctions in the revised manuscript in the discussion (page 9, left column, second paragraph). This should clarify how our findings complement, rather than overlap with, the dynamics explored in extrinsic resetting schemes.

- Section "Results" - "Adaptation and search behavior": The results for the mean first passage times should be explained in more details in order to be clear the comparison to the well-known results in the literature for a Brownian search under resetting or some more generalized search processes under resetting.

We thank the reviewer for emphasizing the importance of comparing our results on mean first-passage times with those in the literature for Brownian motion under resetting. In this work, we focus on numerically investigating the search behavior in self-organized cooperative resetting systems, as described in the "Results" section. While we provide a qualitative argument based on simulations, a full analytical treatment of the mean first-passage times for self-organized resetting dynamics is an exciting direction for future research but is beyond the scope of this study.

To clarify our findings, we compare them to the well-known results for extrinsic resetting. Specifically, we demonstrate that cooperative resetting shares the hallmark feature of extrinsic resetting: the existence of an optimal resetting rate (here, an optimal density ρ) that minimizes the first-passage time for a given target distance. Furthermore, we observe that cooperative resetting generally achieves lower mean first-passage times compared to extrinsic resetting. This improvement arises because cooperative resetting reduces the likelihood of particles remaining near the initial position, increasing the effective flux toward the target and reducing retraction steps close to the goal. This behavior aligns with findings in extrinsic resetting models that incorporate non-resetting windows around the initial position, which similarly reduce first-passage times.

To address the reviewer's comment, we have added these points to the "Results" section (page 6, left column, first paragraph) for clarity and expanded on the broader implications of our findings in the "Discussion" section (page 9, left column, last paragraph).

- Can we give a general conclusions from the obtained results in the manuscript for a general form of the decay of interactions $\mu(\delta)$, for example if mean and/or variance of $\mu(\delta)$ in Eq. (1) exist or not?

Indeed, we can draw general conclusions about the decay of interaction kernels: any interaction kernel that decays slower than δ^{-2} results in localization, whereas kernels decaying faster than δ^{-2} do not. This conclusion arises from the requirement for the normalizability of the probability density function, as we detail on page 4 in the first paragraph. Furthermore, this implies localization does not occur if the integral $\int \delta \mu(\delta) d\delta$ (the mean of μ) exists. We strengthened the emphasis on this insight in the revised manuscript (page 4, left column, 2nd paragraph).

- In the last paragraph before DISCUSSION there is a typo "... (Fig. ??b)". Please correct it.

We corrected this typo.

- Appendix A, relation A1: The partial derivative should be in respect to x . Please correct it.

As for Reviewer #1, we thank the attentive reviewer #2 and would like to apologise for making this typo. This typo is corrected in the revised manuscript.

Reviewer #3 (Remarks to the Author):

In this manuscript, the authors introduce and study a model of "self-organised" resetting which they study (partly) analytically and numerically. In this model of independent Brownian walkers a pair of walkers is reset to its center of mass with a rate that depends on the mutual distance between them. The authors mainly study a one-dimensional situation where the rate is an algebraically decaying function of the inter-particle distance with an exponent α . Interestingly they find markedly different behaviors for $\alpha < 2$ and $\alpha > 2$. For $\alpha > 2$, the diffusion is a stronger than the resetting mechanism and the system does not reach a steady state, while for $\alpha < 2$ there exists a normalisable steady state, where the density profile has a stretched exponential tail. The authors talk about a "localization" transition. Exactly at $\alpha = 2$, they find a similar transition as the density of particles ϱ crosses a critical value $\varrho = \varrho_c$, separating a delocalised phase at low density from a localised one at high density. The authors also study (numerically) the mean first passage time to a target and show that the search is generally improved by self-organized/cooperative resetting compared to standard/extrinsic resetting. Two applications of these concepts are discussed: one in a toy model of sexual interaction in fungi and the other in the context of shared mobility service design.

I find the model quite interesting (although the idea of "self-organized resetting" is not entirely new, see below), leading to a rich phenomenology, despite its simplicity. The paper is well written and I also appreciated that the authors provide a study which combines both analytical (although approximate) arguments, which seem quite reliable, and numerical simulations. In particular, the existence of this localization/delocalization transition is quite interesting and it will certainly trigger further research around such models. Therefore, I would like to recommend the publication of the present manuscript. I would like however to ask the authors to consider the following points:

We thank the reviewer for their positive evaluation of our work and valuable suggestions to further improve the manuscript.

1) As I said, a similar idea of "self-organized" resetting already appeared in the recent literature, for instance:

Majumdar, S. N., Sabhapandit, S., & Schehr, G. (2015). Random walk with random resetting to the maximum position. *Physical Review E*, 92(5), 052126.

Here also, the location of the resetting position is generated by the dynamics and not by an external operator.

We thank the reviewer for highlighting the work of Majumdar et al. (2015), which examines a single random walker reset to its previously realized maximal position. Unlike external resetting, this mechanism is generated by the stochastic process itself. As the reviewer correctly points out, this share similarity to self-organised resetting, yet distinct differences prevail. The process described by Majumdar et al. (2015) is by construction a single-particle non-Markovian dynamics, in difference to our many-particle Markovian dynamics. Notably, Majumdar et al. (2015) elegantly transform this process into a two-dimensional Markovian resetting scheme, where resetting occurs to the $x=0$ axis, resembling an extrinsic resetting process.

We find it insightful to extend the idea of Majumdar et al. (2015) to many-particle dynamics, where resetting occurs to the maximal current position of all particles - which would render this an instance of self-organized resetting. However, such a variation of the original process would require a fundamentally different approach, as Majumdar et al.'s transformation would not map directly to extrinsic resetting in this context.

While we initially decided against referencing Majumdar et al. (2015) in our manuscript, as this publication explicitly considers a one-particle dynamics, putting this mechanism outside the categorization scheme we sketched in Figure 1. Yet, as the reviewer correctly pointed out the similarities to our work, we now acknowledge its relevance and similarity to self-organized resetting. We have added a discussion (page 9, left column, the last two paragraphs) of non-Markovian resetting mechanisms and their connection to our findings citing Majumdar et al. (2015).

2) The authors talk about μ as an interaction, which is a bit misleading, since it is not an interaction in the usual sense in statistical physics (i.e. associated to some energy for instance). This is a rate at which two particles "react". This could be emphasized a bit better (it is said in the present version but a bit hidden I think).

We thank the reviewer for pointing out the potential for confusion regarding the term "interaction" in describing μ . While the term is commonly used in agent-based models

and biophysics to describe such reaction rates, we acknowledge that this usage differs from the conventional interpretation in statistical physics, where interactions are typically associated with energy terms. To enhance the clarity and accessibility of our manuscript, we have now emphasized the precise meaning of "interaction" in the revised version (page 3, left column, first paragraph).

3) Could the author justify a bit more the mean-field approximation they use to solve (3). Does it become exact in the limit $N \rightarrow \infty$? This can be checked numerically for instance. Note that such decoupling is often used in the "hydrodynamic" description of many-particle systems (for instance in the context of the Dean-Kawasaki which the authors could refer to).

We thank the reviewer for highlighting the need to justify the mean-field approximation used to solve equation (3). In deriving our theoretical predictions, our primary focus is on the behavior in the tail of the distribution, as this determines—via the normalization condition—whether the dynamics admit steady-state solutions and localization. To achieve this, we employed a mean-field approximation alongside a series expansion for $\chi \rightarrow \infty$. Consequently, deviations from our predictions are expected near the mean $\langle x \rangle$. This approach provides an accurate description of the asymptotic behavior, though it does not yield a complete solution for the probability distribution.

A notable distinction between our system and stochastic many-body Langevin dynamics often treated in hydrodynamic descriptions (e.g., the Dean-Kawasaki equation) lies in the time-discrete nature of the resetting interactions, which require a different approximation strategy. Interestingly, within the mean-field limit, we derive an equation resembling the McKean-Vlasov equation, if we do not perform the radial integral in Appendix C. However, unlike the mean-field limit of the Dean-Kawasaki equation, our formulation includes an additional correction factor in front of the interaction integral. We have extended the derivation in Appendix C and added a reference to emphasize this connection.

Regarding the asymptotic behavior, we find that our theoretical predictions agree well with numerical simulations, particularly for large systems ($N \rightarrow \infty$), as shown in Figure 2c. We have clarified in the caption of Figure 2c that the value of ϱ was not fitted but directly calculated from the simulation parameters.

4) On p. 3, in Eq. (4) there should be a minus sign on the right hand side.

We thank the reviewer for carefully reading our manuscript. The reviewer is correct that we have been inconsistent in considering the validity of our results both in the limit $\chi \rightarrow \infty$ and $\chi \rightarrow -\infty$. We have corrected Eq. (4).

5) On p. 3, second column: the third paragraph should be part of the second one -- not a new one. As it is, it is a bit misleading because the authors still discuss the $\alpha = 0$ case there.

We thank the reviewer for pointing out this logical flaw. We agree with the reviewer that these two paragraphs should be merged. This is corrected in the revised manuscript.

6) On p. 4, I guess that the authors mean that the density is not normalisable ** for $\varrho < 1$ ** (and not the converse).

The reviewer is correct in pointing out this typesetting error. As described in the subsequent sentence, we find that the density is not normalisable for small resetting rates, and thus $\varrho < 1$.

7) On p. 4, they use once ρ but use otherwise ϱ to denote the mean density.

We thank the reviewer for pointing out this typesetting error. We now consistently use ϱ to denote the density.

8) On p. 4, can one estimate how the rate of dispersion behaves close to ϱ_c . For $\varrho < \varrho_c$, as ϱ approaches ϱ_c , it probably vanishes as $\propto (\varrho_c - \varrho)^\nu$ with some exponent ν . What is this exponent? Does the mean-field prediction gives the correct value?

As the reviewer correctly noted, the change in the spreading rate as a function of the rescaled density ϱ in Figure 2d is indeed reminiscent of critical behavior. We numerically assessed the long-time scaling $\lim_{t \rightarrow \infty} \partial_t \sigma(t)^2$ as a function of $|\varrho_c - \varrho|$ and found power-law dynamics, $\lim_{t \rightarrow \infty} \partial_t \sigma(t)^2 \propto |\varrho_c - \varrho|^\nu$, with an exponent $\nu = 1.234 \pm 0.016$. A simple rational number that is within the confidence bounds is $\nu = 5/2$. This result is illustrated in the attached figure.

Our theory is only valid for the tail of the steady-state probability density function. While this approach effectively determines whether steady states can exist and describes their asymptotic behavior, it is inherently limited in capturing transient dynamics. To fully describe this dynamics of the mean squared displacement one would need to solve a nonlinear second-order partial differential equation that governs the transient behavior. We were unable to find literature that reports exponents with this value in the context of localization phase transitions or spreading phenomena.

In the revised version of the manuscript we now report the value of the critical exponent and added the plot shown below as an inlay to Fig. 2d.